# Multi-User Nonlinear Optical Cryptosystem Based on Polar Decomposition and Fractional Vortex Speckle Patterns

Vinny Cris Mandapati [1,†], Harsh Vardhan [1,†], Shashi Prabhakar [2], Sakshi [3], Ravi Kumar [1,*], Salla Gangi Reddy [1], Ravindra P. Singh [2] and Kehar Singh [4,*]

1   Department of Physics, SRM University—AP, Andhra Pradesh 522502, India;
    gangireddy.s@srmap.edu.in (S.G.R.)
2   Quantum Science and Technology Laboratory, Physical Research Laboratory, Navrangpura,
    Ahmedabad 380009, India
3   Department of Chemical Engineering, Ben-Gurion University of the Negev,
    P.O. Box 653, Beer Sheva 8410501, Israel
4   Optics and Photonics Center, Indian Institute of Technology Delhi, New Delhi 110016, India
*   Correspondence: ravi.k@srmap.edu.in (R.K.); kehars@physics.iitd.ac.in (K.S.)
†   These authors contributed equally to this work.

**Abstract:** In this paper, we propose a new multiuser nonlinear optical cryptosystem using fractional-order vortex speckle (FOVS) patterns as security keys. In conventional optical cryptosystems, mostly random phase masks are used as the security keys which are prone to various attacks such as brute force attack. In the current study, the FOVSs are generated optically by the scattering of the fractional-order vortex beam, known for azimuthal phase and helical wavefronts, through a ground glass diffuser. FOVSs have a remarkable property that makes them almost impossible to replicate. In the input plane, the amplitude image is first phase encoded and then modulated with the FOVS phase mask to obtain the complex image. This complex image is further processed to obtain the encrypted image using the proposed method. Two private security keys are obtained through polar decomposition which enables the multi-user capability in the cryptosystem. The robustness of the proposed method is tested against existing attacks such as the contamination attack and known-plaintext attack. Numerical simulations confirm the validity and feasibility of the proposed method.

**Keywords:** optical security; fractional vortex; polar decomposition; vortex speckles

## 1. Introduction

Recent developments in digital technology and the wide availability of the internet have posed many challenges such as the security of information during transmission and storage. A huge amount of information is transmitted over open channels daily, such as medical images, law enforcement investigations, sensitive intelligence reports, criminal investigations, bank/ATM passwords, etc. The protection of personal information from unauthorized users is one of the major concerns in today's world. Therefore, the development of hybrid security protocols is the need of the hour. In this regard, recently, the optical techniques have drawn considerable attention due to their various advantages over digital counterparts, such as multi-dimensional capabilities, parallel processing, and several degrees of freedoms (i.e., phase, coherence, wavelength, polarization, and orbital angular momentum of light) [1,2]. The first optical cryptosystem, double random-phase encoding (DRPE), was demonstrated in 1995. In DRPE, the two-dimensional image is encoded into a stationary white-noise using two statistically independent random phase masks (RPMs) placed, respectively, in the spatial and Fourier domain [3]. With the passage of time, several other DRPE-based optical cryptosystems were reported in different transform domains [4–8]. However, it was found that due to the inherent linear and symmetric

nature of DRPE architecture, it is vulnerable to various cryptographic attacks, e.g., known-plaintext [9], chosen-plaintext [10], and chosen-ciphertext attacks [11]. To overcome this issue, an asymmetric cryptosystem based on phase truncation in the Fourier domain (PTFT) was proposed [12], but it was found that the PTFT-based asymmetric methods are also susceptible to iterative phase retrieval-based attacks [13]. Some robust encryption methods based on other optical aspects such as interference [14,15], Kronecker product [16], 2D non-separable canonical transforms [17], diffractive imaging [18], polarization [19], phase shifting [20], and digital holography [21], etc., have been proposed.

Asymmetric techniques have attracted the researchers' attention as these techniques, the encryption and decryption keys, are different, which makes it difficult for the attacker to retrieve the information. Several mathematical decompositions have also been integrated with optical approaches to design hybrid cryptosystems, such as equal modulus [22,23], random modulus [24,25], singular value [26], and vector decomposition [27]. Recently, to introduce the multi-user capabilities in optical techniques, polar decomposition-based optical cryptosystems have also been developed [28–30]. Moreover, few of the optical encryption techniques have been demonstrated experimentally [31–37]. An optical cryptosystem based on the single-lens imaging architecture and phase-retrieval algorithm (PRA) was demonstrated experimentally by Mosso et al. [31]. Furthermore, the encryption techniques based on interference [32,33], joint Fresnel-transform correlator with double optical wedges [34], and optical encryption of grayscale information [35], have also been verified experimentally.

In most of these cryptosystems, the computationally generated RPMs are used as the security keys. These patterns are not robust enough against brute force attack if one can use high-performance computing facility. In this paper, we experimentally generate random patterns or security keys using the scattering of fractional-order optical vortex beams. The optical vortex beams are known for having a dark core at the center, which is due to their helical wavefronts. Recently, fractional-order vortices have gained a lot of interest due to their unusual propagation characteristics [38–41]. Here, we change the order, defined as the number of helical wavefronts completed in one wavelength, in the fractional steps for producing RPMs. We studied the sensitivity for decrypting the image by changing the order of the vortex beam up to 0.0001 and found that it is robust against all attacks. The paper is arranged in the following manner: The theoretical background and the proposed method are discussed in Section 2. The results for the encryption, decryption, and sensitivity with respect to various parameters are presented in Section 3, and finally, we conclude in Section 4.

## 2. Theoretical Background and Methodology

### 2.1. Polar Decomposition (PD)

In linear algebra, polar decomposition is used to factorize the matrices into a set of linearly independent parts, including one rotational matrix and two symmetric ones [42,43]. The polar decomposition of a two-dimensional image, $I(x, y)$, of size M × N is as follows:

$$PD \{I(x, y)\} = [R \ U \ V] \tag{1}$$

where R is a rotational matrix and U, V are two symmetric and positive semi-definite matrices. The dimensions of all R, U, and V matrices are the same as that of the input image. To reconstruct the input image, only one of the symmetric matrices and the rotational matrix are required, i.e., $I = R \times U$ or $I = V \times R$. This type of polar decomposition is frequently utilized in continuum mechanics [42]. If I is invertible, then U and V are positive semi-definite matrices, and R is unique. If $I \in P^{n \times n}$, for the left or right $PD$, R is orthogonal $(RR' = I_n)$ and U (or V) is symmetric with non-negative eigenvalues. Polar decomposition has many applications in various fields such as factor analysis, aero-space computations, optimization, matrix square root, etc. [42,43].

### 2.2. Generation of FOVS

We generated the speckle (optical random patterns) by scattering the fractional-order optical vortex beams through a rough surface, i.e., a ground glass plate (GGP). The fractional-order vortex beams were generated by modifying the phase of the incident laser beam using a reflective spatial light modulator (SLM) (Hamamatsu LCOS-SLM, X15223 series) using an optical configuration like that discussed in Ref. [44]. Then, we scatter these beams through a GGP and record the corresponding speckles. The schematic of the experimental setup used to record the FOVS is shown in Figure 1. We used a He-Ne laser (Spectra Physics) of wavelength 632.8 nm as the source, and the CCD camera (FLIR) with pixel size 3.65 μm for recording the speckles. These recorded speckles were used for encrypting and decrypting the image. We tested the sensitivity of the proposed scheme by varying the order in the fractional steps, i.e., negative powers of 10.

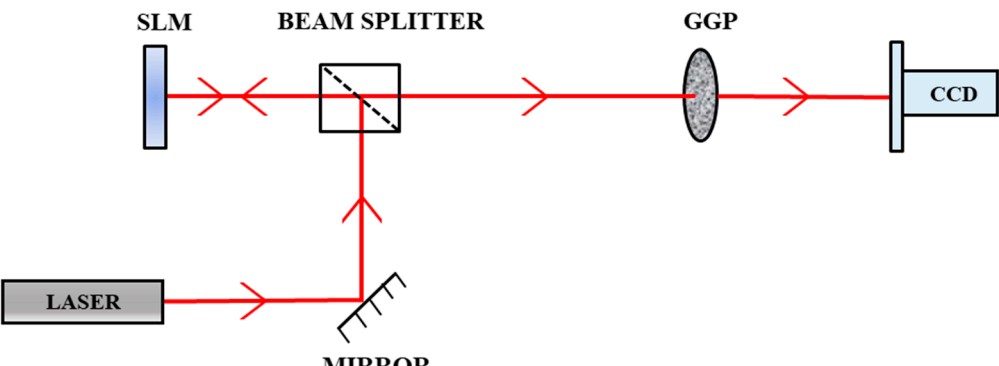

**Figure 1.** Generation of fractional-order vortex speckle pattern. SLM: spatial light modulator; GGP: ground glass diffuser; CCD: charged-coupled device.

### 2.3. Proposed Cryptosystem

In this section, the process of the encryption and decryption is described. The encryption can be carried out by using the following steps:

(a)　First, the input plaintext, $f(x, y)$, is phase encoded as $\exp(i\pi f(x,y))$ and modulated with a fractional optical vortex speckle (FOVS) phase mask.

$$A(x,y) = \exp(i\pi f(x,y)) \times \exp(i2\pi FOVS) \tag{2}$$

$A(x, y)$ is then Fresnel propagated with distance $d_1$ to obtain the complex wavefront $A'(x', y')$, as follows:

$$A'(x',y') = \Im_\lambda^{d_1}[A(x,y)] \tag{3}$$

where $\Im_\lambda^d$ is an operator for the Fresnel propagation to a distance $d$, with wavelength $\lambda$. $(x, y)$ and $(x', y')$ denote, respectively, the coordinates before and after the Fresnel propagation.

(b)　Then, the real and imaginary parts of $A'(x', y')$ are separated, i.e., $re\{A'(x', y')\}$ and $imag\{A'(x', y')\}$. The imaginary part, $imag\{A'(x', y')\}$, is reserved as the first private key and the real part, $re\{A'(x, y)\}$, is further processed using polar decomposition to obtain two more private keys as discussed in Section 2.1.

$$R(x',y') = PD\{re(A'(x',y'))\} = \begin{bmatrix} R & U & V \end{bmatrix} \tag{4}$$

(c)　$R(x', y')$ is then Fresnel propagated to a distance $d_2$ to obtain the complex wavefront $B(x'', y'')$ as follows:

$$B(x'',y'') = \Im_\lambda^{d_2}[R(x',y')] \tag{5}$$

(d) This complex image is further modulated with the amplitude mask FOVS to obtain the final encrypted image, $E(x'', y'')$, as follows:

$$E(x'', y'') = B(x'', y'') \times FOVS \tag{6}$$

The flow charts of the encryption (a) and decryption (b) are shown in Figure 2. The original plaintext can be recovered using all the correct keys and applying the following steps on the encrypted image. First, the ciphertext $E(x'', y'')$ is modulated by the inverse of the FOVS amplitude mask as shown in Equation (6). Then, the output image is Fresnel backpropagated with distance $d_2$, after which the private keys (U or V) are used to obtain the intermediate image, $D_2(x'', y'')$. The first private key, $imag\{A'(x', y')\}$, is then added to the intermediate image and the output is Fresnel back-propagated with distance $d_1$ to obtain the complex image, $D_3(x, y)$. This complex image is further modulated with the conjugate of the FOVS phase mask to obtain the phase image, $D_4(x, y)$. Finally, the angle or phase of this image, $D_4(x, y)$, will give the final decrypted image, $D_5(x, y)$. All these steps can be represented mathematically as follows:

$$D_1(x'', y'') = E(x'', y'') \times FOVS^{-1} \tag{7}$$

$$D_2(x', y') = \Im_\lambda^{-d_2}\{D_1(x'', y'')\} \times U \tag{8}$$

$$D_3(x, y) = \Im_\lambda^{-d_1}[D_2(x', y') + i.imag\{A(x', y')\}] \tag{9}$$

$$D_4(x, y) = D_3(x, y) \times \exp(-i2\pi FOVS) \tag{10}$$

$$D_5(x, y) = \arg\{D_4(x, y)\}/\pi \tag{11}$$

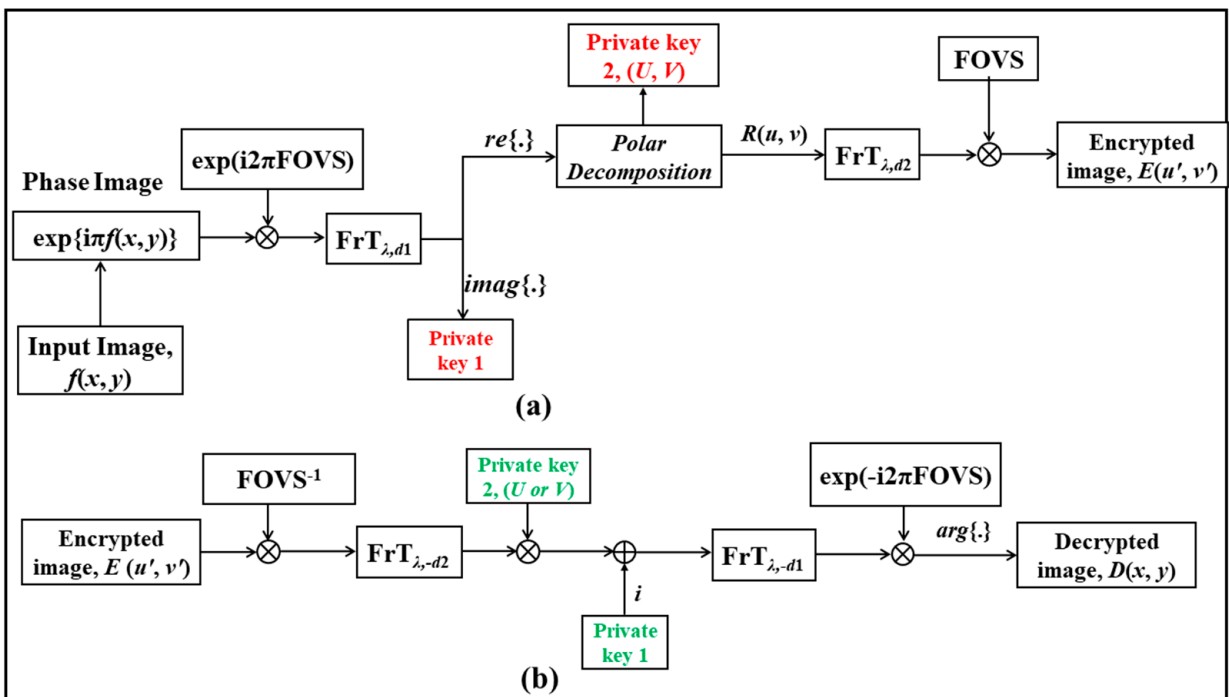

**Figure 2.** Schematic of the proposed technique. (**a**) Encryption and (**b**) decryption processes.

Since the PD and separation of the real and imaginary parts are performed digitally, it is feasible to perform the encryption process digitally. However, the decryption part

can be realized optically by using an optoelectronic setup as in Figure 3. The encrypted image modulated with the inverse FOVS amplitude mask can be displayed on $SLM_1$. A collimated light source can be used for the illuminations. The private key, *U*, is displayed on the $SLM_2$, and the wavefront after that can be combined with the private key 1. The complex wavefront is further modulated by displaying the complex conjugate of the FOVS phase key (i.e., $exp(-i2\pi FOVS)$) on SLM4. Finally, at the output plane, digital holography or a similar system can be employed to record the output field, the amplitude can then be obtained, and the phase part can be calculated by normalizing the output field. The output data are stored in the personal computer (PC). All the SLMs and the CCD are controlled through the PC.

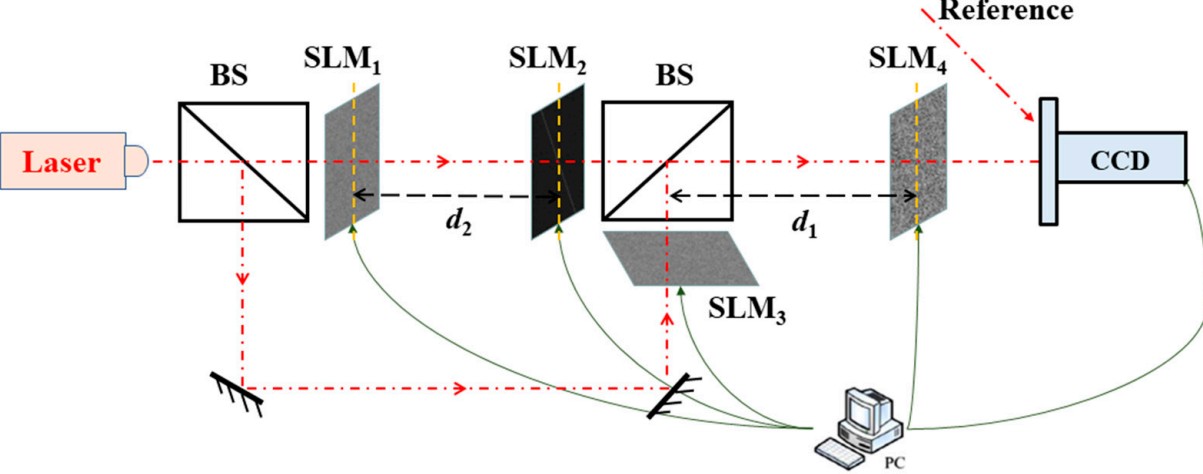

**Figure 3.** Optoelectronic setup for decryption. SLM: spatial light modulator; BS: beam splitter; CCD: charged coupled device; and PC: personal computer.

## 3. Results

The efficacy and validity of the proposed technique is verified through numerical simulations performed using MATLAB$^{TM}$ (version R2022b) on an AMD Ryzen 5 5500U Laptop with 16GB RAM.

### 3.1. Encryption and Decryption Results

The 'Lena' image having $256 \times 256$ pixels, as shown in Figure 4a, is used as the input image for validating our cryptosystem. For the encryption, the original image is first phase-encoded as discussed in the previous section, and then modulated with the FOVS phase mask. The FOVS with order 2.2 is shown in Figure 4b, whereas the first private key, i.e., the imaginary part after the first Fresnel propagation, is shown in Figure 4c. Figure 4d shows the rotational image after polar decomposition, and the two private keys generated after PD are shown, respectively, in Figure 4e,f. The final encrypted image is shown in Figure 4g, whereas the decrypted image with all the correct keys is shown in Figure 4h. For the Fresnel propagation, $d_1$ = 2.5 cm, $d_2$ = 2 cm, and $\lambda$ = 632.8 nm are used for performing the numerical simulations. The FOVS amplitude mask with the fractional order 2.2 is used for modulating the complex wavefront at the final stage. From the results, it can be seen that the encrypted image is like a white random noise and does not reveal any information about the original object. It is to be noted that the PD operation provides a set of two private keys, and for the decryption, only one of them is required. This makes it possible for two different users to have individual private keys at the same time, enabling multi-user capability in the system.

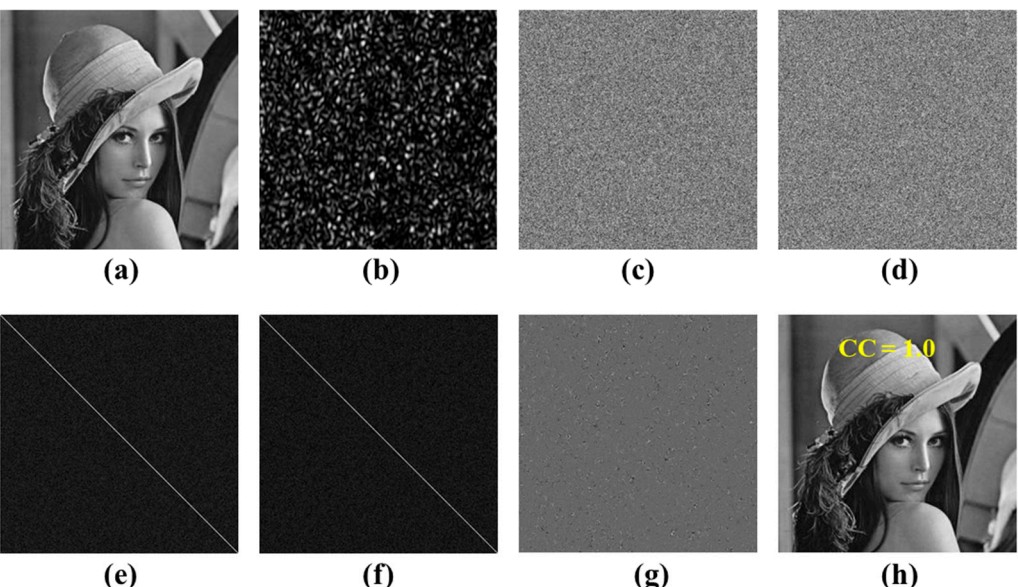

**Figure 4.** (**a**) Original input image; (**b**) FOVS with fractional order 2.2; (**c**) first private key; (**d**) rotational image after PD; (**e**,**f**) private keys generated after PD; (**g**) encrypted image; and (**h**) decrypted image with all correct keys.

### 3.2. Key Sensitivity Analysis

To demonstrate the importance of the security keys, we tested the sensitivity of all the keys by performing the decryption with at least one wrong key, and the corresponding results are shown in Figure 5. For a quality check, the correlation-coefficient (CC) value is calculated between the original image and the decrypted image. The decrypted images with the wrong private key 2 and private key 1 are shown, respectively, in Figure 5a,b. When the wrong Fresnel propagation parameters were used for the decryption, the corresponding decrypted images are shown, respectively, in Figure 5c–e. The propagation distances were changed by 2 mm and the wavelength was changed by 2 nm. To check the sensitivity of the vortex order in FOVS, we generated the speckle patterns with topological charge 2.2, and the corresponding decrypted images with this phase and the amplitude mask are shown, respectively, in Figure 5f,g. The results confirm that to obtain the decrypted image with good quality, all correct keys are required. The low CC values in Figure 5 further validate the importance of all security keys.

The sensitivity to the vortex topological charge and the Fresnel propagation parameters is further checked with very small deviations from the original values. For the FOVSs sensitivity analysis, two sets of FOVSs (one set for the encryption and another for the decryption) with the deviation in the fractional orders (m $\pm$ $\Delta$m) are generated using the fractional orders of the optical vortex beams. The first set of FOVSs has the orders with m = 2, 2.2, 2.22, 2.222, and 2.2222, while the second set of FOVSs is with the orders m = 3, 2.3, 2.23, 2.223, and 2.2223. The sensitivity is checked with the deviation in the fractional order varying from $\pm$1 to $\pm$0.0001. For example, for a deviation of +1, the encryption was carried out with the FOVS corresponding to m = 2, and the decryption was performed with m = 3. Similarly, for a deviation of +0.1, the encryption was carried out with the FOVS corresponding to m = 2.2, and the decryption was performed with m= 2.3, and so on, up to $\Delta$m = 0.0001.

Figure 6 shows the plot of the CC values with the deviations in the fractional order. Figure 7a,b show the sensitivity to, respectively, the deviations in the Fresnel propagation distances $d_1$ and $d_2$, whereas the sensitivity to the deviation in wavelength is shown in Figure 7c. From the plots, it can be clearly seen that the cryptosystem is highly sensitive to the Fresnel propagation parameter and order of the fractional vortex beam, as the CC values are very low for even a small deviation from the original values.

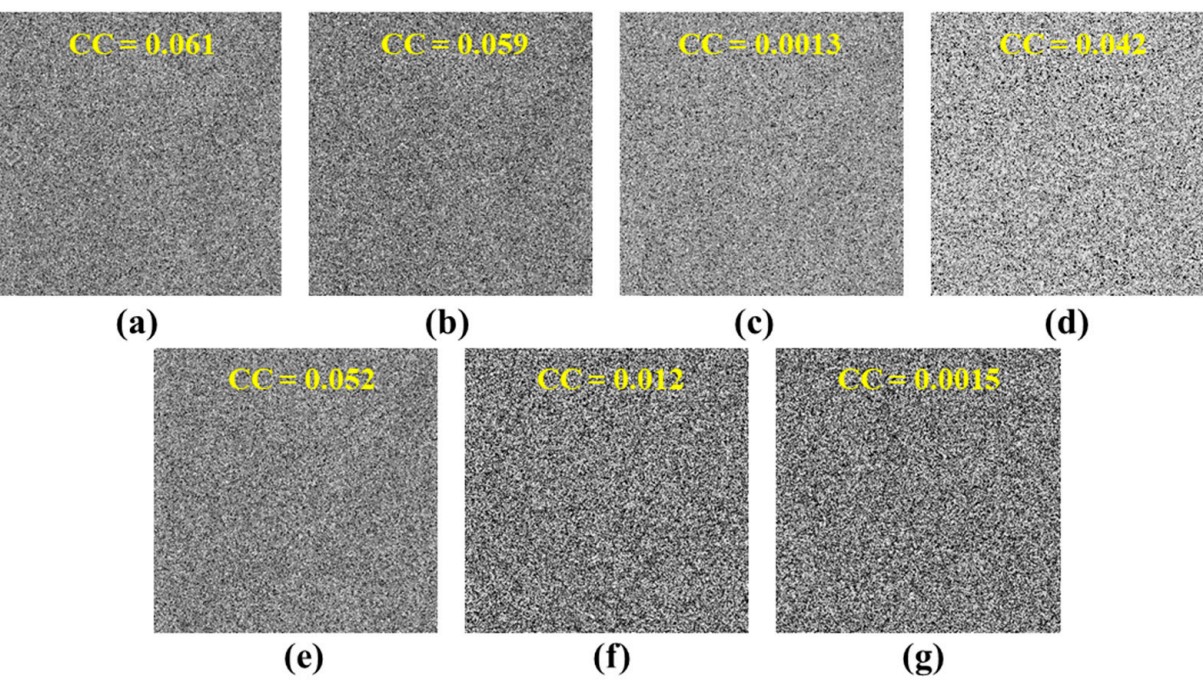

**Figure 5.** Keys' sensitivity analysis. Decrypted image with (**a**) wrong private key 2; (**b**) wrong private key 1; (**c**) wrong d1; (**d**) wrong d2; (**e**) wrong wavelength; (**f**) wrong FOVS amplitude mask; and (**g**) wrong FOVS phase mask.

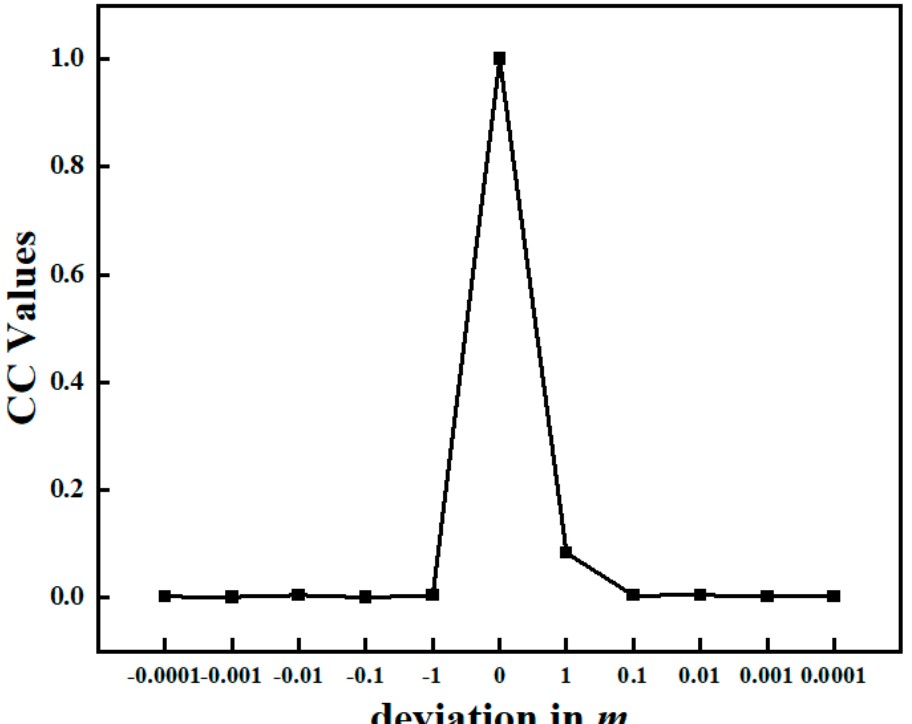

**Figure 6.** Plot of CC values with change in fractional order of the topological charge of vortex beam.

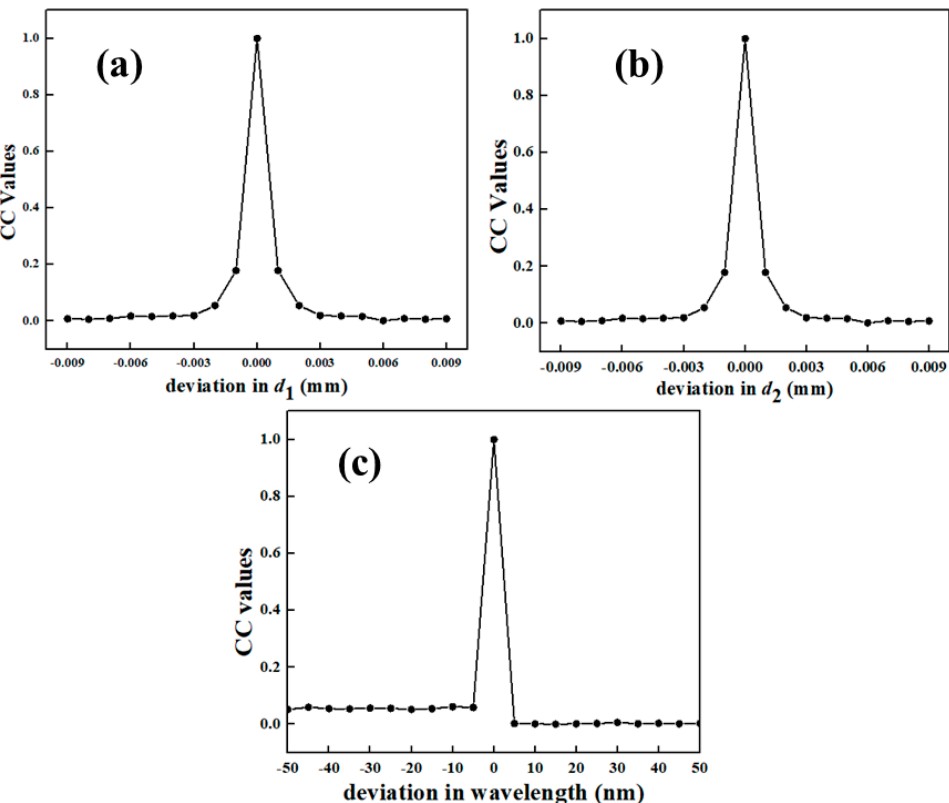

**Figure 7.** Sensitivity of Fresnel propagation parameters. (**a**) CC plot with change in propagation distance $d_1$; (**b**) CC plot with change in propagation distance $d_2$; (**c**) CC plot with change in wavelength.

### 3.3. Attack Analysis

For all new cryptosystems, their robustness against various attacks must be checked to confirm their strength and feasibility for secure information transmission. We checked the robustness of the proposed method against various existing attacks, i.e., contamination attacks (noise and occlusion) and plaintext attacks.

#### 3.3.1. Contamination Attacks

The most common attack during the transmission of secure information is the contamination with unwanted noise or the loss of some part of information. Thus, the robustness of the proposed technique is checked under both possibilities. For the noise contamination test, we contaminated the encrypted image with the Gaussian noise as follows:

$$E' = E(1 + kG), \tag{12}$$

where $E$ and $E'$ are, respectively, the encrypted and noise-contaminated encrypted images. $G$ is the added Gaussian noise with zero mean and 0.05 variance, whereas $k$ represents the strength of the Gaussian noise. The decryption is performed on the noise-contaminated encrypted image with all the correct keys. The results for the noise attack analysis are shown in Figure 8. Figure 8a shows the noise-contaminated image with strength 0.5 and the corresponding decrypted image is shown in Figure 8b. Figure 8e shows the plot of the CC values with noise strength. From the results, it can be seen that the proposed method is robust against noise contamination. Similarly, for the occlusion (information loss), some part of the encrypted image is occluded, i.e., some pixels were assigned zero values. For that, a square section in the central region of the encrypted images corresponding to 5% and 10% pixels are blocked (set to zero), as shown, respectively, in Figure 8c,d. The corresponding decrypted images from the occluded encrypted images are shown in Figure 8f,g. The retrieved images under noise and occlusion contamination reveal significant information

about the original input image, which confirms the robustness of the proposed method against such environment.

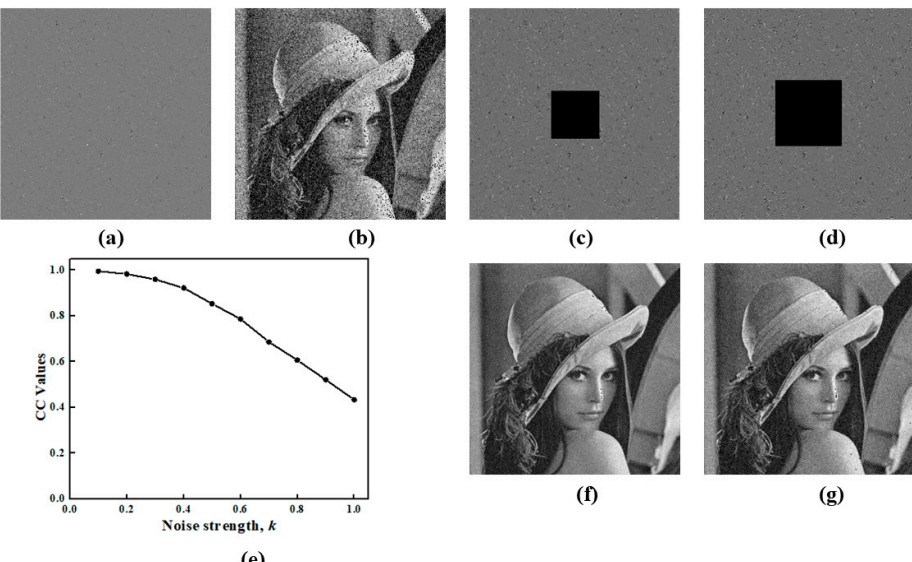

**Figure 8.** Contamination attacks. (**a**) Noise-contaminated encrypted image; (**b**) decrypted image from (**a**); (**c**,**d**) occluded encrypted image with 5% and 10% occlusion; (**e**) CC value plot with noise strength; (**f**,**g**) decrypted images from (**c**,**d**), respectively.

### 3.3.2. Known-Plaintext Attack

We also checked the robustness of the proposed method against the known-plaintext attack (KPA). For plaintext attacks, the intruder already has some information about the cryptosystem, such as knowledge of the input plaintext or ciphertext. In KPA, with partial or complete information about the plaintext–ciphertext pair, the intruder tries to retrieve the private security keys using the iterative algorithms [10–12]. On the other hand, in the case of the CPA, the attacker tries to obtain the security keys using some specific plaintexts (e.g., a series of impulse functions). Additionally, it is to be noted that in case of a KPA, the attackers usually have more information about the cryptosystem than the CPA. In the present study, we tried to retrieve the private key 2 using the KPA and then performed the decryption using the retrieved key. The corresponding results, the plot of the CC values with the iteration number, are shown in Figure 9, and the decrypted image with the retrieved key after 1000 iteration is shown in the inset. The CC values saturate at a very low value near 0.05, which shows the strength of the proposed technique against such attacks.

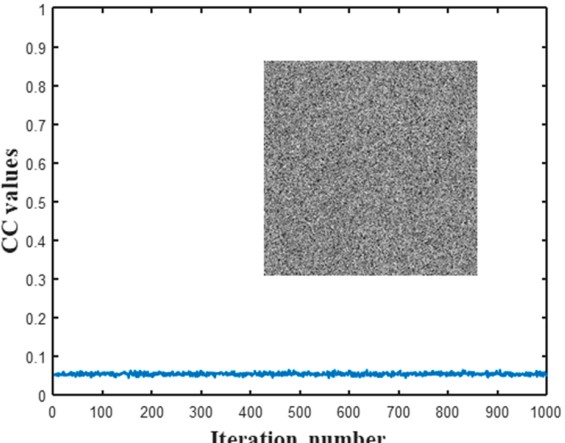

**Figure 9.** Known-plaintext attack result. Plot of CC values versus number of iterations. The decrypted image with the retrieved key after 1000 iteration is shown in the inset.

## 4. Conclusions

In conclusion, we presented a nonlinear optical cryptosystem having multi-user capabilities. The proposed scheme has many security keys which include the FOVS, Fresnel propagation parameters, and two private keys. The physically generated speckles from the vortex beams are very difficult to replicate. The sensitivity of all the keys is discussed. It is evident that the FOVSs are highly sensitive to the fractional order, which enhances the overall security of the system. The robustness of the proposed method is checked against noise-, occlusion-, and known-plaintext attacks. The presented results validate the efficacy of the proposed method and sensitivity to the various security keys. The application of the fractional order and other properties of the vortex beam should be explored more to design sophisticated cryptosystems.

**Author Contributions:** Conceptualization, S.P., R.K., S.G.R., K.S. and R.P.S.; methodology, V.C.M., H.V., S.P., S.G.R., S. and R.K.; software, V.C.M., H.V., S. and R.K.; validation, V.C.M. and H.V.; formal analysis, V.C.M. and S.; investigation, V.C.M., H.V. and S.; resources, S.G.R., S.P. and R.P.S.; data curation, V.C.M. and H.V.; writing—original draft preparation, V.C.M., H.V., R.K. and S.G.R.; writing—review and editing, R.K., S.G.R., K.S. and R.P.S.; visualization, V.C.M. and H.V.; supervision, R.K. and S.G.R.; project administration, R.K. and S.G.R. All authors have read and agreed to the published version of the manuscript.

**Funding:** SGR would like to acknowledge the support from the Science and Engineering Research Board (SERB), the Government of India, under the start-up research grant (Grant No. SERB/SRG/2019/000857), and SRM University—AP for seed research grants under SRMAP/URG/CG/2022-23/006 and SRMAP/URG/E&PP/2022-23/003.

**Institutional Review Board Statement:** Not applicable.

**Data Availability Statement:** The data related to the paper are available from the corresponding authors upon reasonable request.

**Conflicts of Interest:** The authors declare no conflict of interest.

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
