# Peer review of "Multi-User Nonlinear Optical Cryptosystem Based on Polar Decomposition and Fractional Vortex Speckle Patterns"

_photonics, doi:10.3390/photonics10050561_

Round 1
Reviewer 1 Report
Please see attached for my comments.

Reviewer 2 Report
In this paper, the authors investigate a new multi-user asymmetric optical cryptosystem using fractional-order vortex speckle (FOVS) patterns as security keys. In conventional optical cryptosystems, mostly random phase masks are used as security keys, which are subject to various attacks such as brute force attack. In the current study, FOVS are generated optically by scattering a fractional order vortex beam known for azimuthal phase and helical wavefronts through a ground glass diffuser. FOVS have a remarkable property that is almost impossible to reproduce. In the input plane, the amplitude image is first encoded in phase and then modulated with a FOVS phase mask to obtain a complex image. This complex image is further processed to obtain an encrypted image using the proposed method. The two private security keys are obtained using Polar's decompo feature, which provides a multi-user capability in the crypto system. Numerical simulation confirms the correctness and feasibility of the proposed method.
The team of authors presented in this paper are well-known experts in the field of optical cryptography.
I think that the article corresponds to the theme of the journal and can be accepted for publication in the Photonics journal after consideration of the following comments.
1. The authors of the article use optical vortices with a fractional topological charge as the basis of their system, however, in a number of articles it is stated that such vortices can exist only in the initial propagation plane, and after that the vortices acquire an integer topological charge [DOI:10.1103/PhysRevA.102.023516]. How do the authors of the article get around this limitation?
2. In the article, a laser with a wavelength of 632.8 nm was used to generate optical vortices, however, in numerical calculations, a wavelength of 632 nm is used. This needs to be fixed.

Reviewer 3 Report
Two major concerns are as follows:
1, Lack of optical validation does not show the advantage of the so-called optically unreplicated keys. Why, the reason should be given;
2, The private keys are generated in the process of the encryption process, which makes it strongly related to the unique plaintext. That's to say, for different plaintexts, one should have different corresponding private keys, this feature heavily violates the basic purpose of an asymmetric cryptosystem. Though, it's still an extremely difficult issue in the optical field as a comment paper and the replies of that paper conclude (Asymmetric cryptosystem using random binary phase modulation based on mixture retrieval type of Yang-Gu algorithm: comment. Optics Letters 38(20), 4044-4044, 2013. DOI: https://doi.org/10.1364/OL.38.004044.). Here, the suggestion is that you do not highlight the property of ASYMMETRIC.
Reviewer 4 Report
In this paper, the authors proposed and investigated a new optical cryptosystem based on fractional vortex speckle patterns encoding. Modeling showed that when attacked in the form of Gaussian noise, the magnitude of which is equal to the magnitude of the signal, the correlation coefficient of the reconstructed image decreases from 1 to 0.4. But, as shown by the authors, the proposed system is very sensitive to deviations from the given distances d1 and d2 in the optical scheme. If, for example, during decoding, the user makes a mistake in the distance in the scheme by only 3 - 9 nm, then he will no longer restore the original image (see Fig. 6a, b). The work can be published after the authors take into account the comments.
Comments
1) A mathematical expression should be added to determine the FOVS function. It is not clear from the text how the helical phase mφ and speckle noise are combined.
2) In (1) FOVS acts as a phase, and in (5) as an amplitude. But the phase can change from 0 to 2π, and the amplitude can change from 0 to 1. This should be explained.
3) In the signature under Fig. 2, place (a) and (b).
4) In (10), divide D5 by π to get the original image f(x,y) in (1).
5) In line 169 it is written that m=2.3, and in line 176 m=2.2.
6) Line 200-201 says that fractional orders of Laguerre-Gaussian beams are being used. The authors should clarify whether they mean that the Laguerre polynomials in this case have a fractional exponent?
7) Figure 5 shows an incomprehensible СС graph. Figure 5 shows that if the encoding was with m=2.2222 and the decoding was with m=2.2223, then the image cannot be restored. Why then, if encoding at m=2, and decoding at m=2.0001, then the image is restored. It is necessary to explain Fig.5.
Round 2
Reviewer 1 Report
I thank the reviewer for the detailed reply. However, I would still like to point out that the lack of experimental section is a significant degradation of the present manuscript. It is really a pity not to have it.
Also, regarding the reply to comment 3, the authors mention that a small modification of the phase would not induce significant speckle pattern variation, but vortices change would be different. This is again, very intuitive because vortices change is a long range phase change. Therefore, I still believe there is not clear value of using fractional charge vortices rather than ordinary beams.
Reviewer 3 Report
I regret to say that the authors' responses and corrections did not convince me, even if I forget the optical experimental validation.
The authors said the basic purpose or advantage of an asymmetric system is to have a private key and a public key, the readers need to know what is the clear advantage, compared with the symmetric cryptosystem. Furthermore, could you please consider the following situation: I have a plaintext, before we sent it to you we personally and arbitrarily divided it into two parts (maybe one of the two parts includes more info, but the other one has less), now, we claim we build an asymmetric cryptosystem and the part with less info is exactly my private key. Is this ridiculous?
Anyway, I know there are a lot of reports that named the PTFT- or EMD-based optical 'asymmetric' cryptosystem, but it's still hard for me to accept that concept, here, the baseline for me is that you delete the word "asymmetric". there are still enough points you could tell about it, security-enhanced, secret share model, one-time pad model, et al.
Round 3
Reviewer 3 Report
I accept this version.